# Investigation on Microstructure, Nanohardness and Corrosion Response of Laser Cladded Colmonoy-6 Particles on 316L Steel Substrate

**DOI:** 10.3390/ma14206183

**Published:** 2021-10-18

**Authors:** Jeyaprakash Natarajan, Che-Hua Yang, Sundara Subramanian Karuppasamy

**Affiliations:** 1Graduate Institute of Manufacturing Technology, National Taipei University of Technology, Taipei 10608, Taiwan; chyang@ntut.edu.tw (C.-H.Y.); t109569401@ntut.edu.tw (S.S.K.); 2Additive Manufacturing Center for Mass Customization Production, National Taipei University of Technology, Taipei 10608, Taiwan

**Keywords:** colmonoy-6 alloy, laser cladding, nanohardness, corrosion mechanism, roughness

## Abstract

316L steel is predominantly used in manufacturing the components of high-pressure boilers, heat exchangers, aerospace engines, oil and gas refineries, etc. Its notable percentage of chromium offers resistance against corrosion and is mostly implemented in harsh environments. However, long-term exposure to these components in such environments can reduce their corrosion resistance property. Particularly at high temperatures, the oxide film formed on this type of steel reacts with the chloride, sulfides, sulfates, fluorides and forms intermetallic compounds which affect its resistance, followed by failures and losses. This work is focused on investigating the hardness, microstructure and corrosion resistance of the laser cladded Colmonoy-6 particles on the 316L steel substrate. The cladded specimens were dissected into cubic shapes and the microstructure present in the cladded region was effectively analyzed using the FESEM along with the corresponding EDS mapping. For evaluating the hardness of the cladded samples, the nanoindentation technique was performed using the TI980 TriboIndenter and the values were measured. The potentiodynamic polarization curves were plotted for both the substrate and clad samples at 0, 18, 42 and 70 h for revealing the corrosion resistance behavior. In addition, the EIS analysis was carried out to further confirm the resistance offered by the samples. The surface roughness morphology was evaluated after the corrosion process using the laser microscope, and the roughness values were measured and compared with the substrate samples. The result showed that the cladded samples experience greater hardness, lower values of surface roughness and provide better corrosion resistance when compared with substrate samples. This is due to the deposition of precipitates of chromium-rich carbide and borides that enhances the above properties and forms a stable passive film that resists corrosion during the corrosion process.

## 1. Introduction

In recent years, surface modification techniques grabbed the researcher’s attention due to their effectiveness in improving the working life of the critical components. These techniques reduce the need for replacing the components and aids in enhancing the reliability and durability of such parts. It also alters the mechanical properties, and these methods are majorly implemented in treating the worn-out surfaces of components that are employed in severe aggressive environments like nuclear reactors, high-pressure boilers, heat exchangers, etc. where the cost for replacing the parts is very high [1,2,3]. Depending on the area of application, surface modification can be carried out in many ways. Thermal spraying is a type of surface treatment method which can be applicable to a wide range of materials. The major limitation of this technique is that after deposition the sprayed material is reported to experience irregular morphology, thereby reducing the durability of the parts. Moreover, this uneven crystallinity may initiate defects like cracks in the coatings at higher temperatures [4,5].

Physical Vapor Deposition (PVD) is another technique that focuses on developing a thin film of coating on the base material. This film lowers the friction, offers wear resistance and can be suitable for both organic and inorganic substrates. However, this method is not cost-effective, requires skilled professionals, lower rate of deposition and has inadequate bonding behavior between the base and the coated layer [6]. On comparing with other surface modification techniques, laser cladding serves as the most appropriate technique for improving the service life of the parts due to its outstanding characteristics. In this process, the coating material is directly deposited in the molten pool of the substrate which is created by using a high-power laser. The cladded specimen exhibits a better surface finish due to the uniform distribution of the cladding material, excellent metallurgical bonding between the cladding and substrate material, fewer pores in the cladded layer, lower dilution and distortion effect [7,8]. Thus, these characteristics paved the way for this technique to treat the repaired surfaces of the parts in supercritical boilers, aerospace engines, space shuttles, reactor core components and other applications that operates in hostile environments [9,10].

Stainless steels are one among the iron-based alloys that have a wide range of applications. It contains a notable percentage of chromium (~11%) with other alloying elements like Fe, C, Mn, Ti and Mo that makes this steel to possess inherent resistance towards corrosion when compared with plain carbon steels. It also exhibits greater hardness, higher strength, better fabricability, acceptable level of wear and corrosion resistance [11,12]. Stainless steels can be categorized into five classes (ferritic, austenitic, martensitic, duplex and precipitate hardened stainless steel) depends upon the crystalline morphology [13]. Austenitic stainless steels are obtained as a result of phase transition of the ferritic phase in the temperature range between 910–1390 °C. It exhibits a face-centered cubic (FCC) crystal structure representing the γ-iron form called the austenite. This class of steel contributes about 70% to the global market of the stainless-steel industry [14]. Among the different grades of austenitic stainless steels, 316L grade is majorly preferred for manufacturing critical components where the corrosion resistance property is the key aspect. This grade contains a higher amount of chromium (~18%), nickel (~10%) with other elements like C, Mn, Si, S, P, etc. and offers resistance to corrosion and oxidation. It is more suitable for high-temperature applications like heavy water plants, pressurized water reactors, boiler tubes, heat exchangers, etc. [15,16].

Prolonged exposure of this steel in such hostile environments results in the degradation of its material properties followed by the failure of the components. The 316L steel parts in the oil and gas industry experience sliding and erosion wear due to the reaction between the chemical substances and the surface [17]. At high temperatures, the 316L steel is prone to corrosive attacks if the working environment contains chlorides, sulfides and fluorides. These precipitates may initiate the formation of corrosive acids like HCl and H_2_SO_4_ which will damage the 316L steel [18]. In addition, stress corrosion cracks were observed in the 316L steel tubes of high-pressure water reactors due to the effects of the water softening agent. This agent will induce caustic corrosion on the walls of the tube [19]. Thus, the factors like high temperature, high pressure, presence of corrosive salts will initiate the corrosion in 316L steel parts in harsh environments. These corrosion attacks can be overcome by performing laser cladding on the 316L steel parts [20].

Furthermore, the choice of the cladding material plays a significant role in deciding the corrosion resistance characteristics of 316L steel parts. Generally, nickel-based superalloys offer excellent corrosion resistance at elevated pressure and temperature conditions. Colmonoy-6 belongs to the family of nickel-based superalloys that contains relatively higher percentages of chromium and a notable percentage of B, Si, C and Fe elements. This heat-resistant alloy is mostly preferred for enhancing the wear and corrosion resistance in aggressive environments [21]. C.R. Das et al. [22] analyzed the characterization of hardfaced Colmonoy-6 powders on austenitic stainless steel and reported that the Colmonoy-6 coating exhibits higher hardness when compared with the austenitic substrate. Ramasubbu et al. [23] evaluated the dilution effect of the Colmonoy-6 deposits using the Gas Tungsten Arc Welding (GTAW) process and found that the dilution phenomenon significantly influences the hardness and wear resistance in the coated samples. Shi Y et al. [24] investigated the microstructure and wear resistance of laser cladded Colmonoy-6 particles on the stainless-steel substrate. Their results prove that the cladded Colmonoy-6 particles have greatly enhanced the wear resistance of the cladded samples compared to the base material. Kumar et al. [25] analyzed the performance of NiCrB coatings on the 316L steel substrate and reported that this coating has improved the friction coefficient, followed by the wear resistance. Jeyaprakash N et al. [26] further confirm that the Colmonoy-6 coatings have a greater influence on the friction coefficient and wear resistance. Mele et al. [27] evaluated the erosion behavior of the Colmonoy-6 alloy coated on the steel for oil and gas applications. Their results showed that the Colmonoy-6 alloy samples exhibit better resistance towards erosion, thereby improving the microhardness and lowering the surface roughness. Hence, from the above literature, the authors investigated the wear resistance, hardness, erosive behavior of Colmonoy-6 depositions using various coating techniques and proved that the Colmonoy-6 particles will enhance the hardness, wear and erosion resistance of the coated samples. However, the corrosion resistance behavior of laser cladded Colmonoy-6 particles at different time intervals and the corroded morphologies, along with their surface roughness characteristics, have not been analyzed.

This work aimed to investigate the corrosion resistance and the corroded morphologies of the Colmonoy-6 clad samples at various time periods along with microstructure of the clad layer, nanohardness and surface roughness characteristics. The microstructure present in the clad layer is revealed by the FESEM technique with the accelerating voltage of 5.0 kV. The EDS spectra were taken for examining the elemental compositions at the clad layer. The nanoindentation tests were performed on the substrate, interface and clad regions to compare the hardness offered by the clad layer. For investigating the corrosion resistance behavior, both the substrate and cladded samples were immersed in the electrolyte solution made by using NaCl for different time intervals and their performance was analyzed in an electrochemical way. The EIS analysis was further carried out on both samples to evaluate the charge transfer resistance offered by the samples. The corroded morphology was analyzed using the FESEM and the surface roughness was measured after the corrosion process. Thus, the results of both samples were compared and presented.

## 2. Materials and Methods

### 2.1. Materials

Commercially available Colmonoy-6 particles and 316L steel plate were selected as coating material and substrate, respectively and purchased from Well-being Enterprise Co., Ltd, Taipei city, Taiwan. The 316L steel plate was sandblasted for obtaining a roughness of ~7 µm and to produce a good bonding between coating and substrate material. The chemical compositions of the as-received Colmonoy-6 alloy powder and the 316L steel substrate are tabulated in Table 1. The FESEM technique (Model: JEOL 6500F, JEOL Ltd., Boston, MA, USA) is used to reveal the microstructure of the Colmonoy-6 particles and their size distribution (Figure 1). The average size of those particles was found to be 160 ± 10 µm.

### 2.2. Laser Cladding Method

The laser cladding process is an interdisciplinary approach that incorporates CAD/CAM, powder metallurgy, laser processing and control techniques. This process can create a thick film of coatings on the substrate material with a better deposition rate and the created films exhibit a uniform distribution of the cladding material with fewer pores or no pores [28]. For obtaining better metallurgical bonding between the substrate and the cladded layer, the 316L steel samples undergo several processing steps before the cladding process. These samples were rinsed with the acetone solution to remove the surface dust and then sandblasted. The surface roughness after the sandblasting was measured to be ~7 µm. Laser cladding was performed on the sandblasted 316L steel samples. The parameters like laser power, feed rate, diameter of the laser beam and scanning velocity influenced the formed cladded layer. After optimizing these parameters, the laser cladding process was performed [29,30]. In laser cladding, a high-power continuous laser beam from the Ytterbium-doped yttrium aluminum garnet (Yb: YAG Trumpf disk laser-4002) (Geometrix Laser Solution, Mambattu, India) was used to irradiate the surface of the 316L steel samples in an argon (shielding gas) environment. This irradiation tends to the formation of the molten metal pool on the surface where the Colmonoy-6 particles were directly deposited in the molten pool. Figure 2 represents the schematic representation of the laser cladding processing, and the processing parameters were listed in Table 2. Thus, the cladded samples were collected and cut into cubic cross-sections of 10 mm. The cross-sectioned samples went through post-processing steps for the effective revealing of the microstructure present in the cladded layer. These steps include silicon carbide and diamond polishing, followed by chemical etching with the solution made by mixing 15 mL CH_3_COOH, 15 mL HNO_3_, 60 mL HCl, and 15 mL distilled water (Unicorn (Taiwan) Chemicals Pvt., Ltd, Taoyuan, Taiwan). In addition, the nanoindentation test was performed to analyze the hardness offered by the cladded specimens.

### 2.3. Corrosion Analysis

The cladded samples underwent the polarization test to analyze the corrosion resistance of the Colmonoy-6 coatings using the Metrohm Autolab instrument (Metrohm Autolab B.V., 3526 KM Utrecht, The Netherlands). This test was performed in an electrochemical workstation at room temperature conditions. The electrolytic solution was prepared as per the ASTM G44 standards by dissolving 3.5 wt% of sodium chloride (NaCl) in 100 mL of deionized (demineralized) water. This solution mimicked the NaCl content in seawater. In nature, the chloride ions are more aggressive, and these ions are the major reason for producing severe corrosion attacks. These attacks can penetrate the passive film, thereby causing the failure of components via corrosion [31,32].

The corrosion test was performed on both the substrate and the cladded samples at different intervals. Excluding the cladded surface, all other surfaces of the samples were sealed to avoid the leakage of the test solution. The electrochemical station consists of three electrodes where the test sample serves as the working electrode and is held rigid. The reference electrode is made by using silver chloride (AgCl) and the platinum electrode is used as the counter electrode. The circuit potential was said to stabilize and the potentiodynamic polarization curves were recorded within the scanning potential range. From these curves, the corrosion-resistant behavior could be analyzed for both samples. Moreover, the EIS analysis was carried out on both samples to further reveal their charge transfer resistance with the prescribed scan rate and frequency range. The surface roughness was evaluated after the corrosion tests on both samples using the laser microscope (Olympus America Inc., Center Valley, PA, USA) and the structural morphologies of the corroded surface are analyzed with the aid of JEOL 6500F FESEM microscope with EDS mapping (JEOL Ltd, Boston, MA, USA).

## 3. Results and Discussion

### 3.1. Microstructure Examination

The cladded samples were evaluated using the JEOL 6500F FESEM microscope for revealing the microstructures present in the cladded region. Figure 3 represents the FESEM micrographs of the cladded samples obtained at different magnifications. Figure 3a shows the region of the substrate (316L steel) and the clad layer in the samples. The thickness of the clad layer was measured to be 700 µm. The interface layer between the substrate and the clad region is indicated in Figure 3b. At higher magnifications (Figure 3c,d), the morphologies could be clearly investigated. From these figures, it can be inferred that the cladded region exhibits laves phase with the combined effect of dendritic and interdendritic structures [34,35,36].

These structures proved the existence of blocky (dark) and floret-like precipitates in the γ-nickel matrix of the cladded region. The dark precipitates are formed due to the effect of chromium-rich carbides, whereas the floret-shaped structure represents the chromium-rich borides precipitates during the cladding process [37]. The chromium boride precipitates can be of three forms—namely, CrB, Cr_5_B_3_, and Cr_2_B. The CrB is formed directly in the melt pool of 316L steel, and the latter forms are due to the effects of the peritectic reaction [38]. Moreover, some intermetallic compounds were present on the clad surface because of the combined effect of Ni and Si. The same author (Jeyaprakash N et al. [33]) carried out the XRD analysis for laser cladded Colmonoy-6 on 316L steel substrate. They concluded that the peaks in their XRD analysis are due to the presence of Cr-rich carbide and boride precipitates in the γ-nickel matrix. The maximum peak was obtained for γ-nickel followed by interdendritic eutectics ((NiFe)_3_ B) and Cr_5_B_3_. Here also, the Cr-rich carbides and borides were reported in the microstructure of the clad layer. Hence, Jeyaprakash N et al. [33] observation matches this paper’s results and also serves as the supporting information regarding the phase analysis. Figure 4a–e shows the corresponding EDS mapping of the samples. This layered image further proves that the existence of elements like Ni, Si, Fe, and Cr in the cladded layer. This figure also reveals that the higher densities of nickel are followed by chromium, iron and silicon elements. Hence, from the FESEM and EDS studies, the microstructure along with the elements present in the clad region are discussed and reported.

### 3.2. Hardness Evaluation

After the morphological studies, the cladded samples underwent the hardness test to investigate the hardness offered by the cladded samples. Hardness can be defined as the built-in characteristic of a material to resist deformation (plastic deformation). Generally, the 316L steel provided an acceptable level of hardness. But the corrosion attacks in harsh environments made this type degrade their property. By using the Hysitron TI980 TriboIndenter equipment (Bruker, Billerica, MA, USA), the nanoindentation test was performed on the cladded samples. The procedures for making an indentation in the samples are as follows. It combines two processes—namely, loading and unloading. The sample is placed in the specimen holder and the constant load (N) was applied on the surface. In due course, the sample experienced elastic deformation. At the peak load, the sample is unable to bear the load and thus the indentation (plastic phase of deformation) was formed on the sample surface. After the indentation, the unloading process continued gradually. Both the loading and unloading processes were plotted as the load to depth curves for obtaining the indentation depth [39,40].

Here, a constant load of 2000 µN was applied on the cladded samples and the indentation was introduced in the clad surface. The hardness values were calculated by selecting three points in the indentation region. Figure 5a represents the plotted load to depth curves and Figure 5b–d shows the SPM images of indentation obtained at the substrate, interface and the clad region. In general, the higher the depth, the lower the hardness offered [41,42]. From this figure, it can be seen that the substrate (316L steel) experienced a greater value of indentation depth (~70 nm). Hence, the hardness value at the substrate region was very low (3.89 GPa). The sandblasting process influenced the surface quality of the specimen. During the sandblasting process, more heat was released, which accounted for grain refinement on the surface, thereby enhancing the hardness of the sample [43]. Further, after the sandblasting process, the surface became very dense. This dense surface might be the reason for obtaining an increased value of hardness at the substrate [44]. The interface layer exhibited a lower depth value compared to the substrate (~58 nm), thereby increasing its hardness (4.62 GPa). The value of indentation depth at the clad region (~40 nm) was comparatively lower than that of the substrate and interface region. Thus, the clad region encountered a higher hardness value (6.15 GPa). This higher value was due to the formation of Cr-rich carbide and boride precipitates at the clad region. Specifically, the chromium boride provided excellent hardness and also enhanced wear resistance offered by the cladded samples [33,45].

### 3.3. Corrosion Behavior Analysis

The outcomes of the hardness evaluation show that the laser cladded Colmonoy-6 samples exhibit better hardness and resistance to wear compared to that of the 316L steel substrate. To reveal the corrosion resistance behavior of the cladded samples, the corrosion test was performed on both the substrate and clad samples at various time intervals (0, 18, 42 and 70 h) in an electrochemical workstation. This station consists of the test solution (3.5% of NaCl) and after stabilizing the open circuit potential, the potentiodynamic polarization curves were plotted as the Tafel plot. Normally, the 316L steel has an acceptable limit of corrosion resistance property due to its chromium content. On reaction with moisture, an oxide film is formed on its surface owing to the effect of Fe and Cr with O thereby resisting the corrosion. This formed layer has two parts, the upper thin layer of iron oxide and the inner thick layer of chromium oxide, respectively [46,47]. The iron oxide layer will resist corrosion to an extent whereas the chromium oxide layer will provide a better level of corrosion resistance. The mechanism behind the oxide layer formation is given in Equations (1)–(8).
(1)$$2\mathrm{Fe}\to 2{\mathrm{Fe}}^{2+}+2e^{-}$$
(2)$$\frac{1}{2}O_{2}+H_{2}O+2e^{-}\to 2{\mathrm{OH}}^{-}$$
(3)$$2\mathrm{Fe}+\frac{1}{2}O_{2}+H_{2}O\to \mathrm{Fe}{\left(\mathrm{OH}\right)}_{2}\to {\mathrm{Fe}}_{2}O_{3}\cdot {\mathrm{xH}}_{2}O$$
(4)$$2\mathrm{Fe}{\left(\mathrm{OH}\right)}_{2}+\frac{1}{2}O_{2}+H_{2}O\to 2\mathrm{Fe}{\left(\mathrm{OH}\right)}_{3}\to {\mathrm{Fe}}_{2}O_{3}\cdot H_{2}O$$
(5)$$\mathrm{Cr}\to {\mathrm{Cr}}^{3+}+3e^{-}$$
(6)$${\mathrm{Cr}}^{3+}+3H_{2}O⟷\mathrm{Cr}{\left(\mathrm{OH}\right)}_{3}+3H^{+}$$
(7)$$\mathrm{Cr}{\left(\mathrm{OH}\right)}_{3}+\mathrm{Cr}\to {\mathrm{Cr}}_{2}O_{3}+3H^{+}+3e^{-}$$
(8)$$2{\mathrm{Cr}}^{3+}+7H_{2}O\to {\mathrm{Cr}}_{2}O_{7}^{2-}+14H^{+}+6e^{-}$$

Thus, the formed oxide layer on the 316L steel will provide resistance towards corrosion to some extent. The molten salt deposits found in the tubes and tanks in oil and gas industries consist of sulfate, chloride which readily reacts with the H^+^ ions forming corrosive acids that react with the formed oxide film and initiate the corrosion process [48,49]. The Tafel plots for the substrate and cladded samples for 0, 18, 42 and 70 h are represented in Figure 6a,b. From this figure, it can be inferred that there is no proof of active to passive transitions during the process which results in the continuous formation of the oxide film. Also, the potentiodynamic polarization curves show the tendency towards passivation. In general, the lower the current density value, the higher the corrosion resistance offered [50,51]. The current densities and the potential values for both samples are tabulated in Table 3. It can be seen that among the substrate samples, the 42 h sample has a lower value of current density (3.31057838 × 10^−6^ A/cm^2^) compared to the 0, 18 and 70 h samples. The current density values are in an increasing trend for the other three samples and the substrate 0 h sample experiences a higher value of the current density (1.69288259 × 10^−5^ A/cm^2^). On the other hand, all the cladded samples exhibit lower current density values compared to their corresponding substrate samples. Further, the 42 h cladded sample shows a low value (2.70147248 × 10^−7^ A/cm^2^) among the cladded samples and offers higher corrosion resistance compared to the other substrate and clad samples.

By using the FESEM technique, the corroded morphology of the substrate samples at various intervals has been analyzed along with its EDS mapping and represented in Figure 7a–h. From these figures, it is reported that the 42 h substrate sample shows a higher percentage of oxide content (55.02%) with uneven surface morphology. The other substrate samples exhibit a decreasing trend in their oxide content and the values are 21.46%, 20.67% and 17.06% for the 18, 70, 0 h substrate (316L steel) samples. Figure 8a–h shows the FESEM and EDS results for the corroded cladded samples. It can be inferred that during the corrosion process, a stable passive film has been developed on the cladded surface which resists corrosion. This stable film is due to the effect of the precipitated chromium-rich carbides and borides. In addition, the passive film formed on the 42 h clad sample is said to have finite surface morphology and provides excellent resistance towards corrosion. As the time increases, the passive film becomes fragile and starts to leave the surface as patches. Meanwhile, during the cladding process, the 316L steel is irradiated above the austenitic temperature which results in the heterogeneous C concentration thereby increase in the carbon content. Also, higher carbon concentration results in improved resistance against corrosion [52]. After the cladding process, there is an increase in the area fraction of carbides, which in turn increases the carbon content [53]. Further, the increased carbon content can be due to the decomposition of metallic bonds in the alloy during high laser power radiation. Table 4 describes the elemental composition of both the substrate and clad samples which was obtained from the EDS mapping. During the corrosion analysis, a stable oxide film is formed on the clad layer that resists the corrosion. Here, when the Colmonoy-6 clad layer is exposed to the corrosive environment, it forms a strong passive film and provides excellent resistance against corrosion. The film formation is because of the presence of floret-like and dark structures. These structures were due to the chromium-rich carbide and boride precipitates which settle down during the cladding process. The principle behind the passive film formation on the cladded samples is schematically illustrated in Figure 9. Thus, the laser cladded Colmonoy-6 particles enhance the corrosion resistance compared to the 316L steel substrate.

### 3.4. Electrochemical Impedance Spectroscopy (EIS) Evaluation

The results of the corrosion analysis proved that the cladded samples exhibit better resistance to corrosion due to the formation of the stable passive film. To further investigate the resistance, the EIS evaluation was carried out using the Metrohm Autolab instrument (Metrohm Autolab B.V., 3526 KM Utrecht, The Netherlands) on both samples for various time intervals. The outcomes of this spectroscopic analysis were plotted as the Nyquist, bode plots and these plots were fitted using NOVA software (Nova’s Software Services Private Limited, Hyderabad, India). The Nyquist plots for both the substrate and clad samples for 0, 18, 42 and 70 h are shown in Figure 10a,c. From these figures, it can be seen that both the Nyquist plots consist of a single capacitive loop. This is due to the electrochemical reaction that takes place during the process for measuring the resistance dominates the electrode reaction. The real and imaginary components of the complex impedance are plotted within the measured frequency limit [54,55]. The semicircular arc in these plots has a direct relationship with corrosion resistance. Larger the arc’s radius, greater is the corrosion resistance offered by the sample [56,57]. From Figure 10a among the substrate samples, the 42 h sample experiences a larger area of arc radius followed by 18, 70 and 0 h samples. Hence, this sample provides better resistance to corrosion compared to the other three substrate samples. The Nyquist plot obtained for clad samples (Figure 10c) reports that all the cladded samples exhibit a better arc radius compared with their corresponding substrate samples. This is due to the effect of the deposition of Colmonoy-6 particles on the substrate’s surface. Further, the 42 h clad sample shows a maximum arc radius than the other clad and substrate samples. Because of the higher arc radius, this sample offers excellent corrosion resistance among all samples.

The bode plot explains the relationship between the phase shift and the logarithm of the frequency applied. The area covered by the curves of this plot is directly proportional to the corrosion resistance. Greater the covered area, higher is the resistance experienced by the samples [58,59]. On analyzing the bode plot obtained for substrate samples (Figure 10b), the curve of the 42-h substrate sample has the higher covered area thereby providing higher corrosion resistance than the other substrate samples. On the other hand, Figure 10d depicts the bode plots obtained for the cladded samples. Better curves are obtained for the cladded samples compared to the substrate samples and also the 42-h clad sample has a larger area covered under the curve and shows better resistance towards corrosion. The equilibrium circuit for calculating the value of polarization resistance is shown in Figure 10b. This circuit is made by using the combination of polarization resistance (R_P_), constant phase element (CPE) and solution resistance (R_S_). Moreover, the R_P_ value influences the corrosion resistance. The corrosion resistance will increase for higher R_P_ values [60]. Table 5 lists the parameters calculated from the equilibrium circuit for both the substrate and the clad samples. On comparing the R_P_ values of the substrate and clad samples, all the clad samples show higher R_P_ values than its relevant substrate samples at different time intervals. In addition, the R_P_ value of the 42 h clad sample (28,798) is nearly four times of the 42 h substrate sample (6264.1). Thus, by using the EIS analysis, it is found that the laser cladded Colmonoy-6 samples exhibit better resistance towards corrosion than the 316L steel.

### 3.5. Corroded Surface Roughness Characteristics

The Tafel and EIS analysis evaluate the corrosion resistance offered by both the substrate and clad samples in terms of electrochemical aspects. The other key factor that decides the corrosion resistance offered by the sample is the surface roughness. Surface roughness is defined as the measure of irregularities on the surface at the micron level. It is reported that surface roughness greatly influences the corrosion resistance of the samples. During the cladding process, the cladding material is deposited in the molten melt pool of the substrate. Hence after solidification, very minute irregularities are observed in the clad layer. These irregularities are considered as the nucleation sites and they are prone to many types of corrosion attacks because it promotes the growth of nucleation thereby initiating the corrosion [61,62]. In general, the clad layer will exhibit increased surface roughness and lower corrosion resistance due to the formation of nucleation sites. Here, the surface roughness of both the substrate and clad samples after the corrosion process was examined using the LEXT OLS5000 3D Measuring Laser Microscope (Olympus America Inc., Center Valley, PA, USA). It is a rapid, non-contact way for calculating the surface roughness with high precision. It uses the areal method that generates a 3D surface texture having enormous areal data. The captured 3D surface textured is processed with built-in software and the surface roughness values (R_a_) can be obtained directly. The value of surface roughness is in an inverse relationship with the corrosion resistance. Higher corrosion resistance is offered by the samples that have a lower value of surface roughness [63,64]. Figure 11a–d shows the obtained microscopic images for the substrate sample at 0, 18, 42 and 70 h. From this figure, it can be seen that all the substrate sample’s surfaces tend to be very rough which leads to higher roughness values. After the corrosion test of the clad specimens, a passive film (oxide film) is formed on the clad layer. This passive film is said to be stable and resists the corrosion thereby reducing the surface roughness and enhancing the corrosion resistance property. The laser microscopic images for the clad samples at different time intervals are represented in Figure 12a–d and the measured surface roughness values for both substrate and clad samples are tabulated in Table 6. From this table, it is evident that all substrate samples have a higher value of surface roughness which is nearly three times that of the clad samples thereby providing less resistance towards corrosion. On the other hand, the cladded samples exhibit minimum surface roughness and also, the 42 h clad sample has a low value of surface roughness (R_a_ = 3.795 µm) when compared with the other clad and substrate samples. Hence, this study reports that the Colmonoy-6 cladded samples have greater resistance towards corrosion with a minimum value of surface roughness.

## 4. Conclusions

With the aid of the laser cladding technique, the Colmonoy-6 powder particles were evenly deposited on the 316L steel substrate. The cladded surface has been investigated for its properties like morphology, hardness, surface roughness and corrosion resistance using the techniques such as FESEM with EDS spectra, nanoindentation, laser microscopy and EIS. The major outcomes are as follows:The microstructural examination reveals that the clad region consists of the hard laves phase of dendritic and interdendritic structures in the γ-nickel matrix. These structures are due to the existence of dark and floret-like precipitates of chromium-rich carbides and borides which aids in enhancing the properties at the clad region.From the nanoindentation study, it is evident that the cladded region experiences a higher hardness value due to lower indentation depth than the substrate and interface region. Moreover, the higher hardness at the cladded surface is because of the chromium boride precipitates thereby enhances the wear resistance.The results of the Tafel plots for both substrate and clad samples show that the cladded samples have lower current density values than the substrate samples. The 42 h clad sample has the lowest current density and offers excellent resistance towards corrosion compared with other clad and substrate samples. In addition, in the 42 h clad sample, a stable passive film is formed that resists corrosion compared to the other passive films observed by FESEM with the EDS technique.The EIS analysis also proves that the cladded samples have a larger arc radius (Nyquist plot) and greater area-covered curves (bode plot) than the substrate samples. Further, the polarization resistance has been calculated for both samples. Thus, the Nyquist, bode and R_P_ values are higher for cladded samples and in particular, the 42 h clad sample provides greater corrosion resistance due to the maximum value of R_P_ (28,798).The surface roughness measurement also confirms that the laser cladded Colmonoy-6 samples have a minimum value of surface roughness. Also, the 42 h clad sample has a lower value of roughness (R_a_ = 3.795 µm) thereby offering maximum corrosion resistance.

The above studies proved the laser cladded Colmonoy-6 particles exhibit greater hardness, lower surface roughness, better resistance towards corrosion. Hence, the Colmonoy-6 particles can be implemented for the critical components that operate in severe aggressive environments for enhancing their working life and durability.

## Figures and Tables

**Figure 1 materials-14-06183-f001:**
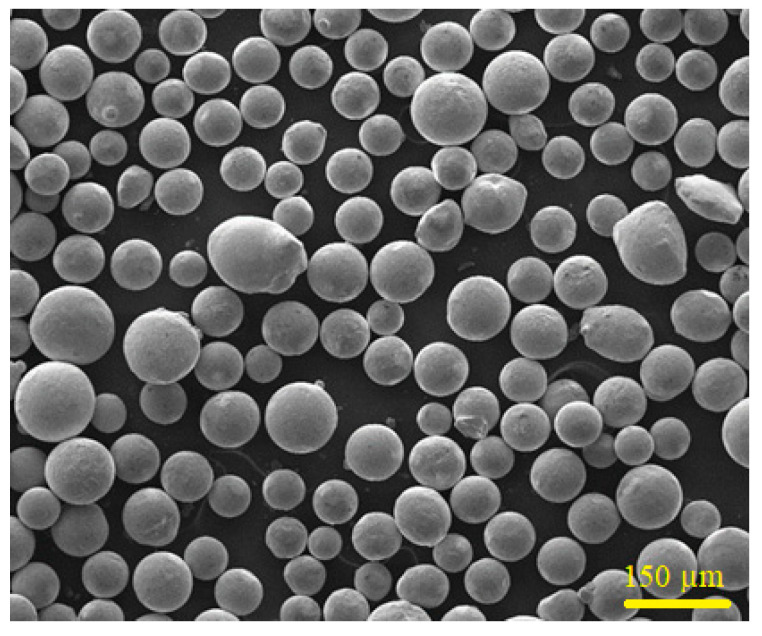
FESEM picture of as-received Colmonoy-6 particles.

**Figure 2 materials-14-06183-f002:**
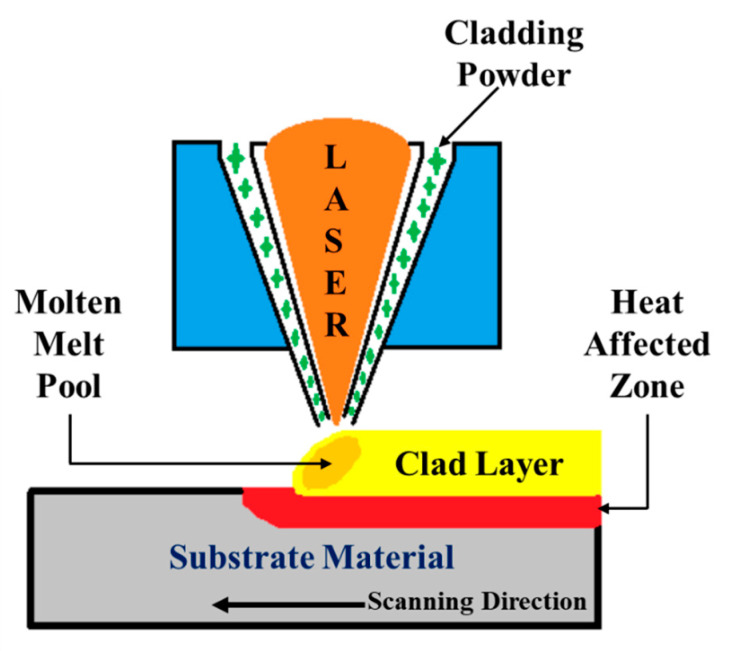
Schematic illustration of the laser cladding process.

**Figure 3 materials-14-06183-f003:**
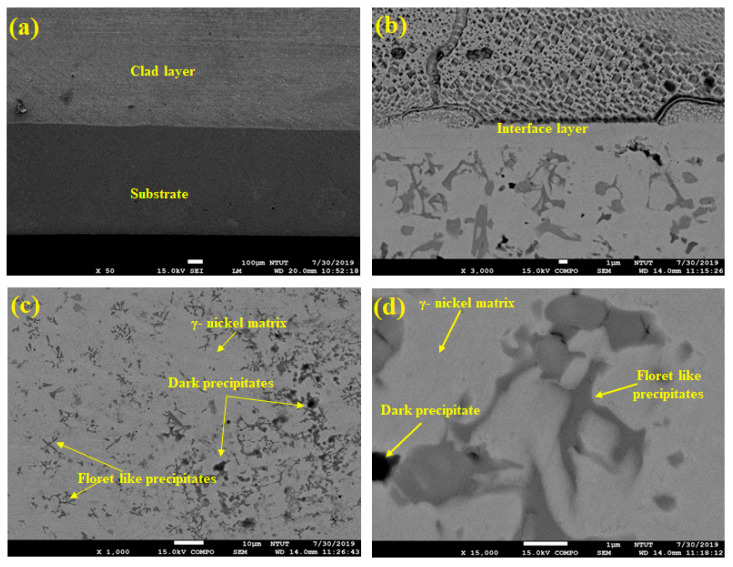
FESEM image of (**a**) Cladded sample, (**b**) Magnified view of the cladded sample, (**c**) Clad region, and (**d**) Magnified view at the clad region representing the dark and floret like precipitates.

**Figure 4 materials-14-06183-f004:**
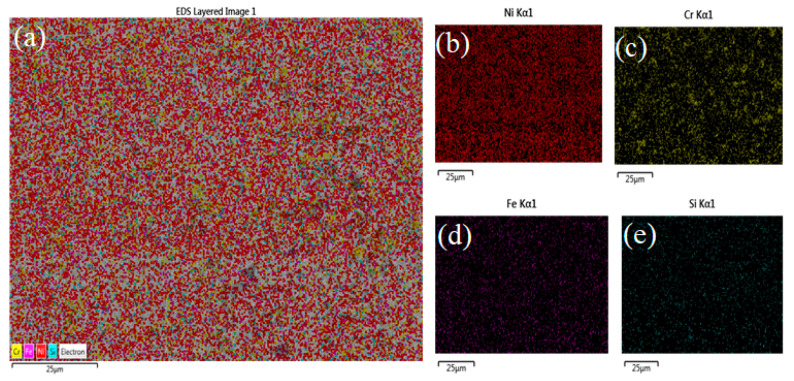
EDS layered image of (**a**) All elements present in the clad region, (**b**) Ni, (**c**) Cr, (**d**) Fe, and (**e**) Si elements.

**Figure 5 materials-14-06183-f005:**
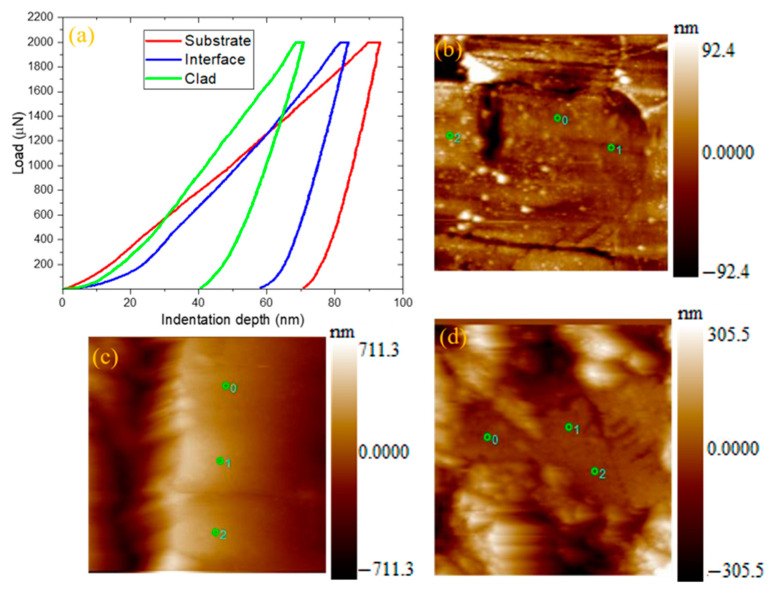
(**a**) Load to depth plot at the substrate, interface and cladded region obtained from the nanoindentation test, SPM images of indentation at (**b**) substrate, (**c**) interface, and (**d**) clad region.

**Figure 6 materials-14-06183-f006:**
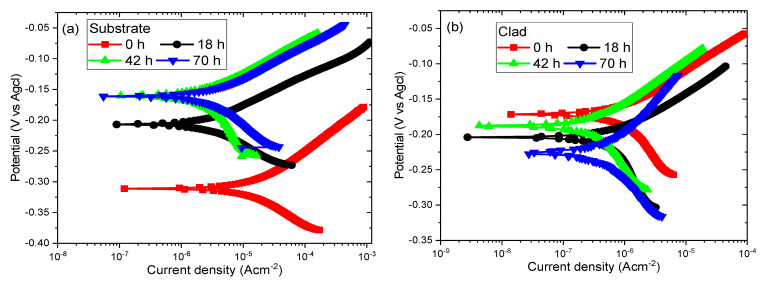
Tafel plot of: (**a**) Substrate (316L steel) and (**b**) Cladded samples at different time intervals.

**Figure 7 materials-14-06183-f007:**
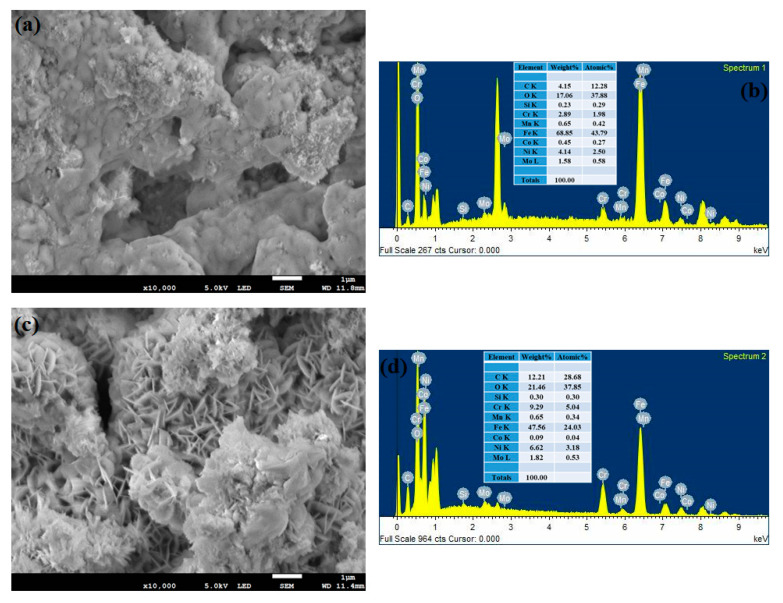
FESEM and EDS spectra of the corroded substrate samples treated at (**a**,**b**) 0 h (**c**,**d**) 18 h, (**e**,**f**) 42 h and (**g**,**h**) 70 h.

**Figure 8 materials-14-06183-f008:**
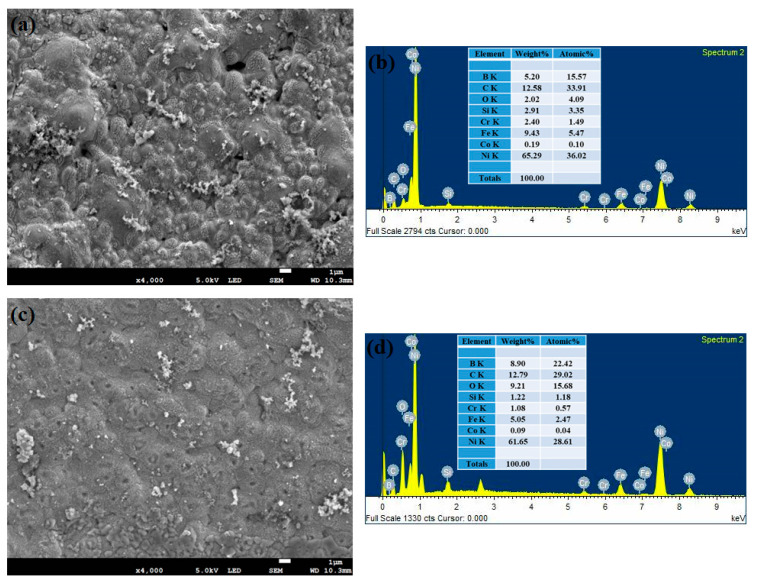
FESEM and EDS spectra of the corroded cladded samples treated at (**a**,**b**) 0 h, (**c**,**d**) 18 h, (**e**,**f**) 42 h and (**g**,**h**) 70 h.

**Figure 9 materials-14-06183-f009:**
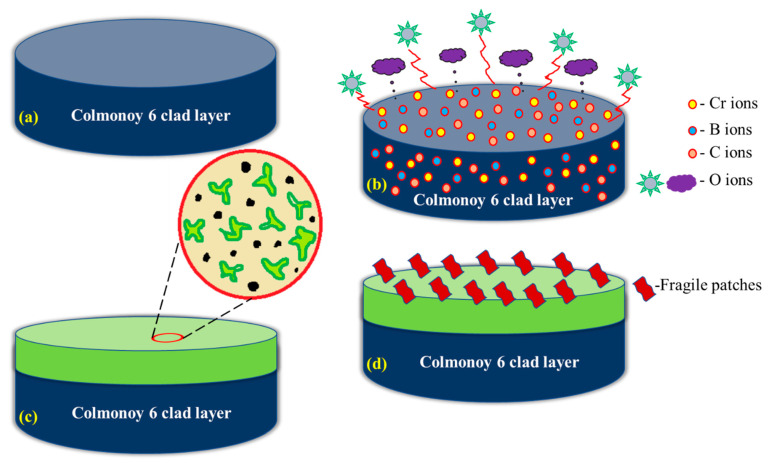
Schematic illustration of passivation layer formed on the clad layer (**a**) Colmonoy-6 clad layer, (**b**) Reaction of Cr, C, B ions present in the Colmonoy-6 with oxygen ions, (**c**) Formation of passive film on the Colmonoy-6 layer which resists corrosion, and (**d**) Fragile passive patches start to leave from the surface.

**Figure 10 materials-14-06183-f010:**
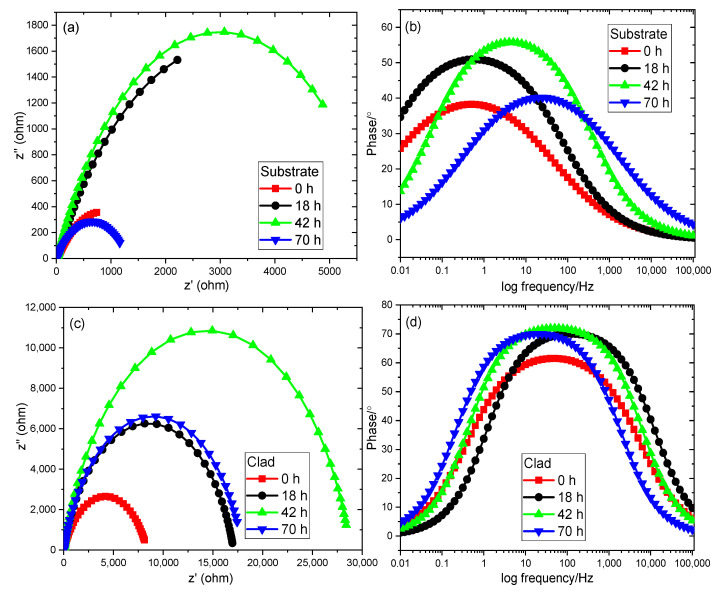
Nyquist and bode plots obtained from EIS analysis for different time treated (**a**,**b**) 316L substrate samples, and (**c**,**d**) Cladded samples.

**Figure 11 materials-14-06183-f011:**
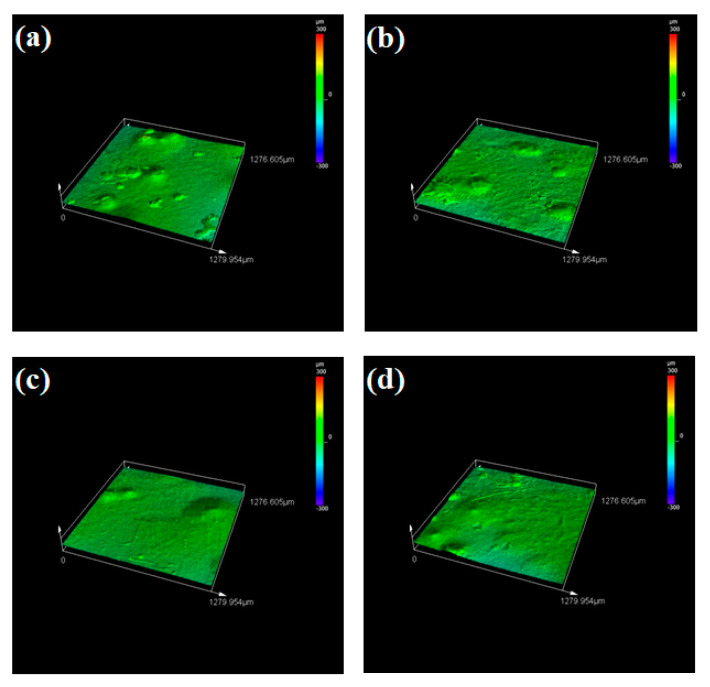
3D surface plots obtained using the laser microscope after the corrosion test for the substrate samples at (**a**) 0 h, (**b**) 18 h, (**c**) 42 h and (**d**) 70 h.

**Figure 12 materials-14-06183-f012:**
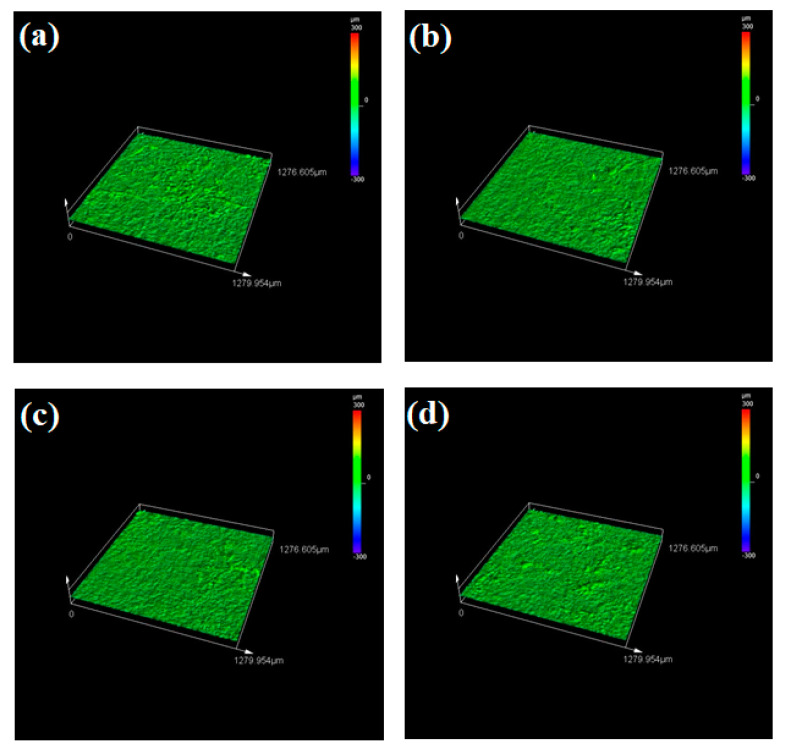
3D surface plots obtained using laser microscope after the corrosion test for the cladded samples at (**a**) 0 h, (**b**) 18 h, (**c**) 42 h and (**d**) 70 h.

**Table 1 materials-14-06183-t001:** Chemical composition of 316L steel and Colmonoy-6 materials.

Material	Ni (%)	P (%)	Cr (%)	Fe (%)	B (%)	Si (%)	C (%)	Mn (%)	S (%)	Mo (%)	N (%)
316L steel substrate	13.00	0.045	18.00	BAL	-	1.00	0.03	2.00	0.015	2.5	0.10
Colmomoy-6	BAL	-	14.3	4.00	3.00	4.25	0.70	-	-	-	-

**Table 2 materials-14-06183-t002:** Processing parameters used in the laser cladding process [33].

Power	Feed Rate	Scanning Speed	Preheat Temperature	Shielding Gas Flow	Carrier Gas Flow
1400 W	9 g/min	600 mm/min	50 °C	25 L/min	6 SD @ 100,000/Pa

**Table 3 materials-14-06183-t003:** Corrosion analysis data.

Duration	E-Current (V)	I-Current (A/cm^2^)
Substrate 0 h	−0.311714771	1.69288259 × 10^−5^
Substrate 18 h	−0.162945671	3.70289779 × 10^−6^
Substrate 42 h	−0.20528438	3.31057838 × 10^−6^
Substrate 70 h	−0.161990662	4.43903655 × 10^−6^
Clad 0 h	−0.171866157	1.00896469 × 10^−6^
Clad 18 h	−0.203961517	7.21036311 × 10^−7^
Clad 42 h	−0.188327674	2.70147248 × 10^−7^
Clad 70 h	−0.226584607	5.16628422 × 10^−7^

**Table 4 materials-14-06183-t004:** The elemental compositions of both substrate and clad samples at different time intervals.

Specimens	Element		C K	O K	Si K	Cr K	Mn K	Fe K	Co K	Ni K	B K	Mo L	Totals
Substrate 0 h		Weight%	4.15	17.06	0.23	2.89	0.65	68.85	0.45	4.14	-	1.58	100.00
		Atomic%	12.28	37.88	0.29	1.98	0.42	43.79	0.27	2.50	-	0.58	
Substrate 18 h		Weight%	12.21	21.46	0.30	9.29	0.65	47.56	0.09	6.62	-	1.82	100.00
		Atomic%	28.68	37.85	0.30	5.04	0.34	24.03	0.04	3.18	-	0.53	
Substrate 42 h		Weight%	8.00	55.02	−0.04	2.08	−0.01	32.27	0.50	0.96	-	1.23	100.00
		Atomic%	13.99	72.27	−0.03	0.84	−0.01	12.14	0.18	0.34	-	0.27	
Substrate 70 h		Weight%	8.46	20.67	0.19	9.81	1.46	56.01	-	3.43	-	−0.03	100.00
		Atomic%	21.47	39.41	0.20	5.75	0.81	30.59	-	1.78	-	−0.01	
Clad 0 h		Weight%	12.58	2.02	2.91	2.40	-	9.43	0.19	65.29	5.20	-	100.00
		Atomic%	33.91	4.09	3.35	1.49	-	5.47	0.10	36.02	15.57	-	
Clad 18 h		Weight%	12.79	9.21	1.22	1.08	-	5.05	0.09	61.65	8.90	-	100.00
		Atomic%	29.02	15.68	1.18	0.57	-	2.47	0.04	28.61	22.42	-	
Clad 42 h		Weight%	28.69	5.68	9.64	4.05	-	12.01	0.40	208.76	25.85	-	295.08
		Atomic%	25.59	3.80	3.68	0.83	-	2.30	0.07	38.10	25.62	-	
Clad 70 h		Weight%	9.16	21.12	2.81	1.45	-	10.61	−0.19	44.44	10.60	-	100.00
		Atomic%	18.44	31.93	2.42	0.67	-	4.59	−0.08	18.31	23.71	-	

**Table 5 materials-14-06183-t005:** The elemental compositions of both substrate and clad samples at different time intervals.

Duration	Element	Rs	R_P_	CPE	
	Parameter	R	R	Y_0_	N
Substrate 0 h (χ^2^ = 0.83517)	Value	13.259	1279.4	0.00053777	0.53608
	Estimated Error (%)	4.061	4.323	5.601	1.865
Substrate 18 h (χ^2^ = 0.92959)	Value	15.576	5968.7	0.00035181	0.67476
	Estimated Error (%)	2.839	5.864	3.757	1.246
Substrate 42 h (χ^2^ = 0.49617)	Value	15.531	6264.1	0.0014249	0.6167
	Estimated Error (%)	1.867	12.253	2.329	1.123
Substrate 70 h (χ^2^ = 0.6928)	Value	18.121	1708.5	0.003173	0.51266
	Estimated Error (%)	2.408	14.564	3.687	2.199
Clad 0 h (χ^2^ =0.86605)	Value	8.5646	8373	0.000061818	0.71986
	Estimated Error (%)	4.105	3.429	4.309	0.928
Clad 18 h (χ^2^ =0.97353)	Value	9.4141	17069	0.000011745	0.80688
	Estimated Error (%)	4.901	2.922	4.996	0.850
Clad 42 h (χ^2^ = 0.8678)	Value	10.683	28798	0.00001558	0.82306
	Estimated Error (%)	3.826	3.048	4.028	0.740
Clad 70 h (χ^2^ = 1.2022)	Value	11.18	17984	0.000044658	0.80815
	Estimated Error (%)	3.678	4.086	4.491	0.958

**Table 6 materials-14-06183-t006:** Roughness values of substrate and cladded samples after corrosion test.

Specimens	Roughness Value—R_a_ (µm)
Substrate—0 h	15.475
Substrate—18 h	13.133
Substrate—42 h	10.745
Substrate—70 h	14.239
Clad—0 h	5.359
Clad—18 h	4.954
Clad—42 h	3.795
Clad—70 h	4.273

## Data Availability

The experimental datasets obtained from this research work and then the analyzed results during the current study are available from the corresponding author on reasonable request.

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
