# Peer review of "Investigation on Microstructure, Nanohardness and Corrosion Response of Laser Cladded Colmonoy-6 Particles on 316L Steel Substrate"

_materials, 2021, doi:10.3390/ma14206183_

Round 1
Reviewer 1 Report
1) It is necessary to clearly show in the introduction what is the novelty and originality of this work.
2) It is necessary to clarify what protective gas was used during laser cladding.
3) It is necessary to specify which equipment (laboratory or industrial, its brand) was used for laser cladding. The same should be done with respect to corrosion tests, microstructure and surface analysis.
4) The phase analysis should be more accurate. At least, the XRD data should be given. Taking into account these data, it is necessary to make changes to the conclusions.
5) The authors are recommended to try another etching technique, which allows more clearly identifying the features of the microstructure of the cladded layer. For example, the etching technique used in the following recent work: R.A. Savrai et al, The structural characteristics and contact loading behavior of gas powder laser clad CoNiCrW coating, Optics and Laser Technology, 2020, Vol. 126, Art. 106079, https://doi.org/10.1016/j.optlastec.2020.106079
6) The hardness of the steel substrate is quite large (3.89 GPa). It is necessary to analyze the reasons for this (the influence of sandblasting is possible).
7) Table 3: What are the reasons for giving 8-9 decimal places for the potential and current density? It is necessary to compare these data with the measurement error.
8) The statement about the significant effect of roughness on corrosion resistance requires clarification. Does this mean that if the substrate is polished, its corrosion resistance will significantly improve?
Reviewer 2 Report
The subject matter is very interesting, important, and has a special value considering practical applications. However, there are still some things that could be improved, and a few questions that have to be answered before publication. Therefore, I suggest a mandatory revision of the following points to increase the quality of the paper:
1. SS should be removed before marking 316L steel
2. Authors should provide the names of all used test devices.
3. There are no parameters of SEM, for example, accelerating voltage that influence on the EDS microanalysis resolution.
4. With which reagent the coatings were etched?
5. The authors have described the phases in the microstructure. They should carry out XRD tests. In addition, they should perform chemical composition studies (eg EDS) at characteristic phases in the microstructure which they have described in their manuscript.
6. The authors should show SEM pictures of the places where the hardness tests were performed. Tests at such low loads are erroneous and concern very small areas of the microstructure. The authors should perform hardness tests with higher loads.
7. Why did the authors perform the corrosion resistance in a 3.5% NaCl solution? It requires comments.
8. The EDS analysis is not suitable for the determination of carbon and boron content. The authors approached this uncritically. Why was there such a high carbon content on the surface of the samples after the corrosion resistance test, since Colmonoy alloy and 316L steel have carbon content of 0.7% and 0.03%, respectively?
9. The authors examined the surface roughness. However, they did not provide any information about roughness parameters that are commonly used, eg Ra.
10. Fig.9 should be described in more detail in the text.
11. The authors should show the linear distribution of element concentrations on the cross-section of the clad layer and the substrate, both before and after the corrosion process.
This manuscript in the presented form is not acceptable for publication in the Materials. The major revision is necessary.
Round 2
Reviewer 1 Report
The revised version of the manuscript is acceptable.
Reviewer 2 Report
The authors did not take into account all comments of the reviewer but cleared up all doubts. Some experiments were unfeasible due to the pandemic. In my opinion, this does not detract from the quality of the manuscript. This manuscript in the presented form is acceptable for publication in the Materials.